# Visible Photocatalytic Hydrogen Evolution by g-C_3_N_4_/SrZrO_3_ Heterostructure Material

**DOI:** 10.3390/nano13060977

**Published:** 2023-03-08

**Authors:** Shizhao Si, Yanfei Fan, Dan Liang, Ping Chen, Guanwei Cui, Bo Tang

**Affiliations:** College of Chemistry, Chemical Engineering and Materials Science, Collaborative Innovation Center of Functionalized Probes for Chemical Imaging in Universities of Shandong, Key Laboratory of Molecular and Nano Probes, Ministry of Education, Shandong Normal University, Jinan 250014, China

**Keywords:** photocatalytic, hydrogen evolution, strontium zirconate, graphitic carbon nitride, heterostructure

## Abstract

A heterostructure material g-C_3_N_4_/SrZrO_3_ was simply prepared by grinding and heating the mixture of SrZrO_3_ and g-C_3_N_4_. The morphology and structure of the synthesized photocatalysts were determined by scanning electron microscopy (SEM), X-ray diffraction (XRD), X-ray photoelectron spectroscopy (XPS), energy-dispersive X-ray spectroscopy (EDS), high-resolution transmission electron microscopy (HRTEM) and infrared spectra. It showed visible light absorption ability and much higher photocatalytic activity than that of pristine g-C_3_N_4_ or SrZrO_3_. Under the optimal reaction conditions, the hydrogen production efficiency is 1222 μmol·g^−1^·h^−1^ and 34 μmol·g^−1^·h^−1^ under ultraviolet light irradiation and visible light irradiation, respectively. It is attributed to the higher separation efficiency of photogenerated electrons and holes between the cooperation of g-C_3_N_4_ and SrZrO_3_, which is demonstrated by photocurrent measurements.

## 1. Introduction

Hydrogen evolution by photocatalytic water splitting is proposed to be the most promising approach to address energy shortages and environmental pollution caused by the overuse of fossil fuels. Inorganic semiconductors are widely used as photocatalysts, such as TiO_2_, ZnO, titanate, tantalate, etc [1,2,3]. Perovskite-type SrMO_3_ (M = Ti, V, Zr and Nb) has attracted a lot of attention from researchers due to its remarkable magnetic and electronic transport properties [4,5,6,7,8,9]. SrZrO_3_ is a typical perovskite structure material, which has been widely used as fluorescent materials, hydrogen sensors, proton conductors, refractory materials, et al. [10,11,12,13]. It has an energy band structure suitable for the photocatalytic decomposition of water with a band gap of 5.25 eV. However, due to the wide energy gap, it can only respond to ultraviolet light, resulting in a low efficiency of water splitting by photocatalysis. Ion doping or coupling with other materials are the main methods to expand the spectral response range and improve photocatalytic efficiency. Torres-Martínez et al. found that when 1% CuO was loaded onto the surface of SrZrO_3_, the efficiency of hydrogen evolution was significantly increased under UV light [14]. Zhou et al. synthesized a new MoS_2_/SrZrO_3_ photocatalyst heterojunction, which was applied to photocatalytic hydrogen evolution under UV light irradiation, and found that the heterostructure with a content of 0.05 wt% MoS_2_ exhibited a high H_2_ release rate of 5.31 mmol h^−1^ [15].

Graphitic carbon nitride (g-C_3_N_4_) is a new non-metallic polymer material that has attracted much attention due to its high thermal and chemical stability, as well as excellent electrical properties. Especially in the field of photocatalysis, g-C_3_N_4_ has visible light absorption ability and photostability. It has been widely applied in photocatalytic hydrogen evolution, water oxidation, degradation of organic pollutants, and photosynthesis. [16,17,18] Significantly, carbon nitride itself is a photocatalyst with a narrow energy gap. It can be used as a visible light-absorbing composition in heterogeneous structures by coupling with other semiconductor materials [19,20].

Herein, to expand the spectral response range and improve the photocatalytic efficiency of SrZrO_3_, a simple heterostructure material g-C_3_N_4_/SrZrO_3_ was prepared by grinding and heating the mixture of SrZrO_3_ with g-C_3_N_4_. To the best of our knowledge, this is the first report on the photocatalytic activity of a hybrid heterostructure using a non-metallic polymer material coupled to SrZrO_3_. Our focus is to explore the g-C_3_N_4_/SrZrO_3_ photocatalytic properties and related influencing factors such as the structural, electronic and optical properties by optimizing the ratio of mixture. A complete characterization of the semiconductors and the heterostructure is presented and photocatalytic mechanisms based on UV-Vis diffuse reflectance spectra and photoelectrochemical measurements are discussed. It showed visible light absorption ability and much higher photocatalytic activity than that of pristine g-C_3_N_4_ or SrZrO_3_.

## 2. Materials and Methods

### 2.1. Materials

Sr(NO_3_)_3_, KOH and urea were purchased from China National Pharmaceutical Group Chemical Testing Co., Ltd. (Beijing, China). ZrOCl_2_·8H_2_O, methanol and ethanol were purchased from Shanghai Macklin Biochemical Co., Ltd (Shanghai, China). All the chemicals are AR grade. The water used in the experiment is secondary deionized water.

### 2.2. Preparation of g-C_3_N_4_/SrZrO_3_

SrZrO_3_ was synthesized by a hydrothermal synthesis method [15]. Typically, Sr(NO_3_)_3_ (1.16 g) and ZrOCl·8H_2_O (1.61 g) were dissolved in KOH solution (60 mL, 1.0 mol·L^−1^) and stirred for 1 h at room temperature. The obtained mixture was transferred to a polytetrafluoroethylene reaction kettle and heated at 200 °C for 24 h, then naturally cooled to room temperature. The resulting solid product was filtered and sequentially washed several times with distilled water, dilute acetic acid, and ethanol, and then dried at 60 °C for 12 h. 10.00 g of urea was heated in a high-temperature tube furnace in an air atmosphere at a heating rate of 8 °C/min and kept at 550 °C for 4 h. Then it was naturally cooled down to room temperature and ground into powder.

The SrZrO_3_ and g-C_3_N_4_ samples prepared above were uniformly ground in a mortar in different proportions (20:1, 15:1, 10:1, 5:1, 3:1, 1:1, 1:2) and calcined in a muffle furnace at a heating rate of 5 °C/min for 2 h at 500 °C. Then, it was naturally cooled to room temperature. Finally, a yellowish substance was obtained. Herein, the optimal ratio of SrZrO_3_ and g-C_3_N_4_ for photocatalysis is 10:1.

### 2.3. Photocurrent Measurements

Photocurrent measurement was carried out on an electrochemical analyzer (CHI660D Instruments, Shanghai Chenhua Instrument Co., Ltd., Shanghai, China.) by a standard three-electrode system with 0.1 M NaClO_4_ aqueous solution as electrolyte. The working electrode was prepared as follows: 0.05 g of the sample was ground into a slurry with 0.10 g terpinol. Then, it was coated onto a 4 cm × 1 cm Indium Tin Oxide-coated glass (ITO glass) electrode by doctor blade technique, dried in an oven, then calcined at 290 °C for 30 min under Ar conditions. The as-prepared samples, Pt sheet and saturated calomel electrode were used as working electrodes, the counter electrode and the reference electrode, respectively. Prior to photocurrent measurements, the electrolyte (0.1 M NaClO_4_, pH = 6.56) was purged with Ar for 30 min. A 300 W Xe lamp was used as the light source to measure the photocurrent density by periodic irradiation.

### 2.4. Photocatalytic Hydrogen Production

The photocatalytic hydrogen production test was carried out in an XPA-7 photocatalytic reactor. In a typical process, 10.0 mg of the as-prepared photocatalysts and 10.0 mL aqueous solution containing 20% methanol were mixed in a 20 mL Quartz bottle sealed with a silicone rubber septum. Prior to the photocatalysis experiment, the sample solution was thoroughly deaerated by evacuation and purged with nitrogen for 10 min. Then a 1000 W Xe lamp with an optical filter (cutoff of 420 nm) simulating visible light or a 500 W Hg lamp simulating ultraviolet light was irradiated for 8 h at room temperature under constant stirring. The generated gas was analyzed by gas chromatography (FULI 9750, TCD, Nitrogen as the carrier gas, and 5 Å molecular sieve column).

### 2.5. Characterization of Samples

The morphology and structure of the synthesized photocatalysts were determined by scanning electron microscopy (SEM, SU8010, Hitachi, Ltd. Tokyo, Japan), X-ray diffraction (XRD, D8 Bruker, Bruker, Karlsruhe, Germany), X-ray photoelectron spectroscopy (XPS, ESCALAB 250, Thermo Fisher Scientific Co., Ltd. Waltham, MA, USA), energy-dispersive X-ray spectroscopy (EDS, Oxford EDS, Oxford Instrument Technology Co., Ltd. Oxford, UK), high-resolution transmission electron microscopy (HRTEM, JEM-2020, JEOL (Beijing) Co., Ltd. Bejing, China) and other characterization techniques.

## 3. Results and Discussion

The as-prepared SrZrO_3_ particles exhibit uniform flower-shaped morphology with sizes of about 14 μm composed of some polyhedral microcrystals radiating from a common point (Figure 1a,b). The as-prepared g-C_3_N_4_ exhibits irregular particle morphology with sizes of 200–500 nm (Figure 1c). After they were ground and calcined together, the latter was uniformly dispersed on the surface of the former, resulting in the formation of g-C_3_N_4_/SrZrO_3_ heterostructure material. The morphology of SrZrO_3_ did not change significantly after being modified with g-C_3_N_4_ (Figure 1d). The XRD peaks centered at 2θ = 30.8°, 44.07°, 54.78°, 64.12°, 72.86°, 81.21°, 89.3° of g-C_3_N_4_/SrZrO_3_ is ascribed to (121), (202), (042), (242), (161), (044), (244) crystal planes of perovskite SrZrO_3_ with a orthorhombic phase structure (JCPDS:44-0161). Although pure g-C_3_N_4_ has two peaks at 13.08° and 27.4°, no significant g-C_3_N_4_ peak was observed on the XRD pattern of g-C_3_N_4_/SrZrO_3_ due to the low loading (Figure 1e). The HRTEM image of g-C_3_N_4_/SrZrO_3_ (Appendix A) shows the presence of two different crystal lattices in the composite. The lattice width of 0.290 nm is ascribed to (121) plane of SrZrO_3_, and 0.275 nm to the (200) plane of g-C_3_N_4_, indicating the presence of g-C_3_N_4_ on the surface of SrZrO_3_ [21,22], which is further confirmed in the IR spectra (Figure 1d). Infrared spectroscopy can be used to detect the information of chemical bonds or functional groups contained in the molecule. From the infrared spectrum of g-C_3_N_4_/SrZrO_3_, we can see that 1640 cm^−1^ corresponds to the stretching vibration of CN, 1240 cm^−1^, 1321 cm^−1^, 1412 cm^−1^ correspond to the stretching vibration of aromatic CN, 809 cm^−1^ is the out-of-plane bending vibration of the CN heterocycle, and the peak near 3140 cm^−1^ corresponds to the aromatic ring stretching vibration of the terminal NH_2_ or NH group at the defect site, which is consistent with the results reported in the literature [23]. 500 cm^−1^ corresponds to the stretching vibration of Zr-O, indicating that g-C_3_N_4_ has been successfully combined with SrZrO_3_ to form a g-C_3_N_4_/SrZrO_3_ composite material. We can synthesize the composite catalyst by this method.

The EDS-mapping images show that the g-C_3_N_4_ nanoparticles were evenly distributed on the SrZrO_3_ micron-scale particles as well as nanometer scale (Figure 2 and Appendix A). The chemical states of Sr, Zr, O, C, and N elements in the as-prepared g-C_3_N_4_/SrZrO_3_ materials were determined by XPS spectra (Figure 3a). Two peaks centered at 132.9 eV and 134.6 eV were ascribed to Sr^2+^ 3d_5/2_ and Sr^2+^ 3d_3/2_ of SrZrO_3_, respectively (Figure 3b) [24,25]. Two peaks centered at 181.2 eV and 183.7 eV were ascribed to Zr^2+^ 3d_5/2_ and Zr^2+^ 3d_3/2_ of SrZrO_3_, respectively (Figure 3c). Three C 1s peaks centered at 284.7 eV, 286.1 eV and 288.9 eV were ascribed to C-C, C=O and C=N, respectively (Figure 3d) [26]. Three N 1s peaks centered at 398.8 eV, 400.2 eV and 401.3 eV were attributed to pyridine nitrogen, pyrrolic nitrogen and graphitic nitrogen, respectively (Figure 3e) [27]. Two O 1s peaks centered at 529.4 eV and 531.5 eV were ascribed to adsorbed oxygen and lattice oxygen, respectively (Figure 3f) [28,29].

The photocatalytic activity of the as-prepared catalyst was investigated in a photocatalytic water-splitting process using methanol as a sacrificial agent, which was performed under a 1000 W xenon lamp light source with a cutoff of 420 nm or 500 W Hg lamp irradiation for 8 h. Compared with SrZrO_3_ and g-C_3_N_4_, the g-C_3_N_4_/SrZrO_3_ composite exhibited higher hydrogen production performance whether under ultraviolet light or visible light irradiation (Figure 4a,b). It was found the SrZrO_3_/g-C_3_N_4_ sample with ratio of 10:1 showed the highest photocatalytic activity (Appendix A). Under the optimal reaction conditions, the hydrogen production efficiency is 1222 μmol·g^−1^·h^−1^ and 34 μmol·g^−1^·h^−1^ under ultraviolet light irradiation and visible light irradiation, which it has achieved higher activity compared with previous work (Appendix A), respectively. The g-C_3_N_4_/SrZrO_3_ samples retained photocatalytic activity after four photocatalysis cycles. This indicates that it has excellent chemical stability (Figure 4d).

The as-prepared g-C_3_N_4_/SrZrO_3_ exhibits much higher and broader spectral absorption, including visible light, compared to pristine SrZrO_3_ and g-C_3_N_4_ (Figure 5a). The energy gap of the sample was obtained by plotting (Ahv)_2_ against hv based on the UV-Vis diffuse reflectance data of the catalyst and extrapolating the straight line to the intersection of the scale lines (Appendix A). The energy gap was calculated to be 5.25 eV for SrZrO_3_ and 2.9 eV for g-C_3_N_4_. Combined with the XPS valence band spectrum test (Appendix A), the top of the valence band of SrZrO_3_ is located 2.31 eV below the Fermi level, so the bottom of the conduction band is located at −2.94 eV; the top of the valence band of g-C_3_N_4_ is located at an energy level of 2.33 eV below the Fermi level, so the bottom of the conduction band is at −0.57 eV.

As represented in Figure 5b, in the heterostructure g-C_3_N_4_/SrZrO_3_, both semiconductors are excited under UV–vis light irradiation. After electrons and holes are generated in the semiconductors, since the conduction band position of SrZrO_3_ is more negative than that of g-C_3_N_4_, the photogenerated electrons on the conduction band of SrZrO_3_ can be easily transferred to the conduction band of g-C_3_N_4_, while the more positive holes in the valence band of g-C_3_N_4_ can transfer to the valence band of SrZrO_3_. In the water splitting reaction, electrons on the conduction band of g-C_3_N_4_ and holes in the valence band of SrZrO_3_ achieve the reduction and oxidation of water. Benefiting from the characteristics of spatially separated electrons and holes in heterostructures, the g-C_3_N_4_/SrZrO_3_ materials improving the separation efficiency of photogenerated electrons and holes. Meanwhile, the g-C_3_N_4_ part can also photogenerate independently electrons and holes under visible light irradition, which also contributes to the number of photogenerated carriers. The photocurrent density of g-C_3_N_4_/SrZrO_3_ is much higher than that of pristine SrZrO_3_ and g-C_3_N_4_ under Xe lamp light irradiation (Figure 5c), which indicates that the charge separation in g-C_3_N_4_/SrZrO_3_ is significantly enhanced, resulting in higher photocatalytic activity.

In conclusion, the g-C_3_N_4_/SrZrO_3_ heterostructure reduces the electron-hole recombination, increasing the photo-excited electron density and the separation efficiency of electrons and holes for the photocatalytic water-splitting, thus enhancing the photocatalytic activity.

## 4. Conclusions

In summary, a heterostructure material g-C_3_N_4_/SrZrO_3_ is prepared by grinding the mixture of SrZrO_3_ and g-C_3_N_4_. The as-prepared photocatalyst showed a hydrogen evolution rate of 1222 μmol·g^−1^ h^−1^ under ultraviolet light, which is higher than that of either g-C_3_N_4_ or SrZrO_3_. It is attributed to the higher separation efficiency of photogenerated electrons and holes between the cooperation of g-C_3_N_4_ and SrZrO_3_.

## Figures and Tables

**Figure 1 nanomaterials-13-00977-f001:**
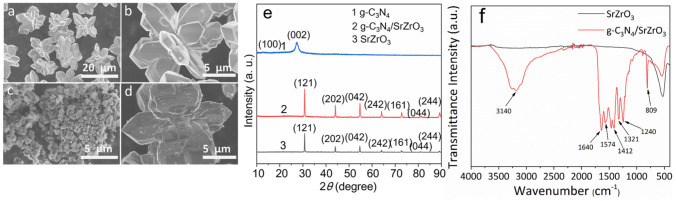
SEM patterns of SrZrO_3_ (**a**,**b**), g−C_3_N_4_ (**c**), and g−C_3_N_4_/SrZrO_3_ (**d**). XRD patterns of SrZrO_3_, g−C_3_N_4_ and g−C_3_N_4_/SrZrO_3_ (**e**). Infrared spectra of SrZrO_3_ and g−C_3_N_4_/SrZrO_3_ (**f**).

**Figure 2 nanomaterials-13-00977-f002:**
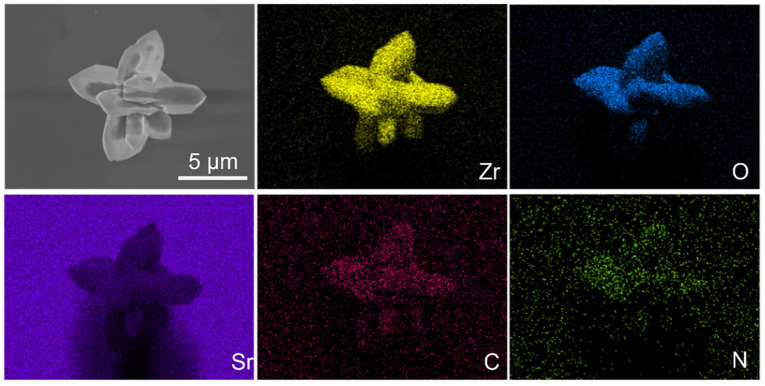
EDS-mapping of g-C_3_N_4_/SrZrO_3_.

**Figure 3 nanomaterials-13-00977-f003:**
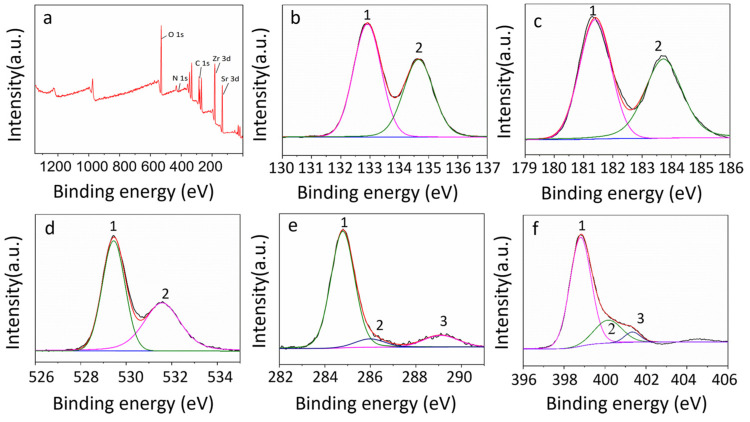
XPS of g−C_3_N_4_/SrZrO_3_, survey XPS spectra (**a**), Sr 3d (**b**), Zr 3d (**c**), O 1s (**d**), C 1s (**e**), N 1s (**f**).

**Figure 4 nanomaterials-13-00977-f004:**
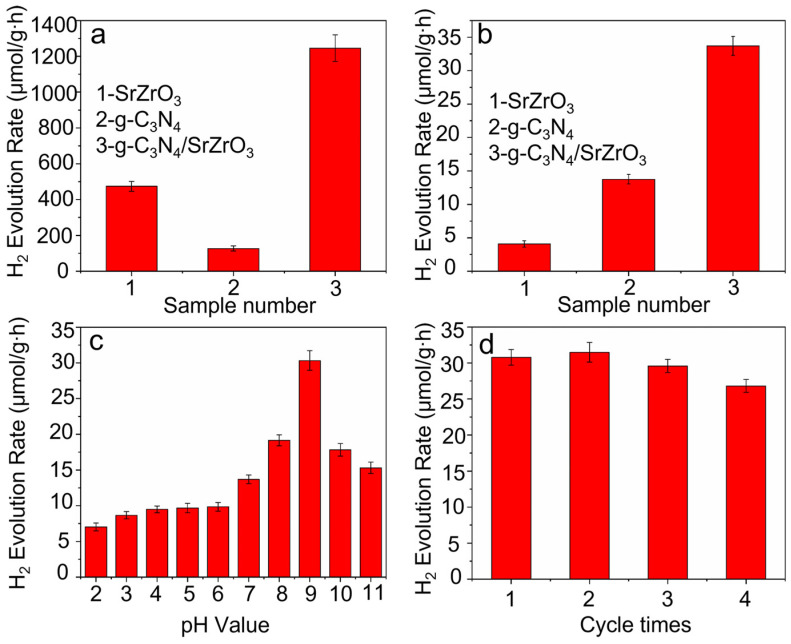
Hydrogen evolution of SrZrO_3_, g-C_3_N_4_, and g-C_3_N_4_/SrZrO_3_ under ultraviolet light irradiation (**a**), visible light irradiation (**b**), different pH values (**c**), and stability test of H_2_ evolution (evacuation every 8 h) for g-C_3_N_4_/SrZrO_3_ under visible light irradiation (**d**).

**Figure 5 nanomaterials-13-00977-f005:**
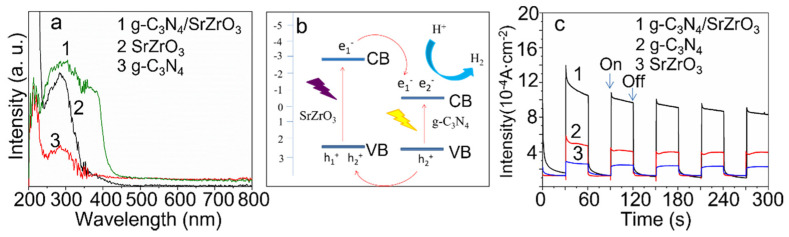
UV-Vis diffuse reflectance spectra of SrZrO_3_, g−C_3_N_4_ and g−C_3_N_4_/SrZrO_3_ (**a**). Photocatalytic mechanism diagram (**b**). Photocurrent of SrZrO_3_, g−C_3_N_4_ and g−C_3_N_4_/SrZrO_3_ (**c**).

## Data Availability

The raw data supporting the conclusion of this article will be made available by the authors, without undue reservation.

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
