# Peer review of "Visible Photocatalytic Hydrogen Evolution by g-C3N4/SrZrO3 Heterostructure Material"

_nanomaterials, 2023, doi:10.3390/nano13060977_

Round 1

Reviewer 1 Report (Previous Reviewer 1)

Although the authors responded to most of the comments, after reviewing the answers, I consider that the manuscript could not be accepted in its present form. The authors were advised to improve the state-of-the-art. The introduction of a new paragraph does not improve the whole section. Also, the reporting style was requested to be changed (at this moment it looks like a technical report, not like a scientific one) and it seems that the authors did not considered necessary. As a consequence, the manuscript cannot be considered for publication in its present form.   

Author Response

Thanks for the reviewer’s valuable suggestion. In the introduction, it has been explained for the purpose of this work and related research progress with concise language. The format of the paper conforms to the requirements of this journal.

Reviewer 2 Report (Previous Reviewer 2)

Accept

Author Response

Thanks for the reviewer’s valuable suggestion. 

Reviewer 3 Report (New Reviewer)

The manuscript describes synthesis of composite g-C3N4/SrZrO3 for the photocatalytic hydrogen production under UV- and visible light. In general, the subject of the work is quite interesting, but the article itself is written rather carelessly, and there is practically no analytical discussion of the results obtained.

Main commens:

1. The relevance and scientific significance of the work should be clearly justified in the introduction. Now the introduction looks sparse.

2. Characterization paragraph should be added to the chapter 2. Materials and Methods.

3. Fig. 1e. All peaks must be clearly marked.

4. It is better to do TEM mapping at nanometer scale.

5. The text and figures from Supporting should be moved to the main text. A more detailed discussion of the characterization results should be made.

6. Fig. 4. The error of measurements should be shown.

7. The activities obtained in the research looks quite low. The table with comparison of activities with recently published data on this kind of systems should be provided.

8. The mechanism of charge separation and electron transfer must be clearly and reasonably explained.

Author Response

1 The relevance and scientific significance of the work should be clearly justified in the introduction. Now the introduction looks sparse.

Per the reviewer’s suggestions, the introduction has been revised and supplemented (line 52-53, line 56-62).

2 Characterization paragraph should be added to the chapter 2. Materials and Methods.

Per the reviewer’s suggestions, the characterization paragraph has been revised and supplemented (line 108-113).

  1. Fig. 1e. All peaks must be clearly marked.

All the peaks have been marked in the Figure 1e.

  1. It is better to do TEM mapping at nanometer scale.

Per the reviewer’s suggestions, the TEM mapping at nanometer scale has been supplemented (line 144, Figure S2).

  1. The text and figures from Supporting should be moved to the main text. A more detailed discussion of the characterization results should be made.

We have revised the text and relevant figures in the SI moving to the main text according to the reviewer's suggestions (Figure 1f, line 130-139).

  1. Fig. 4. The error of measurements should be shown.

All the error of measurements has been supplemented in the Figure 4.

  1. The activities obtained in the research looks quite low. The table with comparison of activities with recently published data on this kind of systems should be provided.

Per the reviewer’s suggestions, the table has been supplemented (line 167-168, Table S1). We have collected recent literature and found that strontium zirconate materials have relatively few applications in photocatalysis research, especially in the field of photocatalytic water-splitting for hydrogen production. Among the existing achievements, we have achieved great performance, expanded the application of perovskite-like strontium zirconate materials, and provided new ideas for the design and development of strontium zirconate materials.

  1. The mechanism of charge separation and electron transfer must be clearly and reasonably explained.

Per the reviewer’s suggestions, the mechanism has been revised and supplemented (line 186-188, line 190-197, line 201-204).

Round 2

Reviewer 1 Report (Previous Reviewer 1)

The manuscript can be accepted in its present from. 

Reviewer 3 Report (New Reviewer)

I'm satisfied with the review.

This manuscript is a resubmission of an earlier submission. The following is a list of the peer review reports and author responses from that submission.

Round 1

Reviewer 1 Report

The article describes visible photocatalytic hydrogen evolution by g-C3N4/SrZrO3 Heterostructure material.

The theme is interesting, although some concerns must be addressed:

-          The abstract is too short and it does not summarize the findings of the study.

-          The state-of-the-art must be improved, at this moment it looks like a technical report, not like a scientific one.

-          What are the aim and novelty of this study? 

-          Full information about used reagents and instruments (company, city, country) must be provided.

-          All figures must be included after their first mention in the text.

-          “ml” must be replaced with “mL” throughout the entire manuscript.

-          Figure 3: “Sampe” must be probably “Sample”?????

-          Please specify whether the used materials were more efficient compared with other studies.

-          The “Results and discussion” section must be improved. At this moment, it looks like a technical report, the same situation as the “Introduction” section.  

-          The “Conclusions” section is mandatory to be improved.

-          It is mandatory that the English language is revised by a native English speaker.

Reviewer 2 Report

1. In the Abstract, the preparation was simply described as "grinding", but according to the manuscript, at least heating is a necessary process, more accurate and detailed description is needed here.

2. Line 67 in 2.2, what is the ratio of the SrZrO3 and g-C3N4?

3. It is suggested to move Figure S3 and S4 from the SI to the main body, which could better characterize the products.

4. I would like to suggest going through the manuscript more carefully for clarity, syntax and correctness. The English should be improved for the sake of clarity.

Reviewer 3 Report

Review comments

The research article on " Visible Photocatalytic Hydrogen Evolution by g-C3N4/SrZrO3 Heterostructure Material" reports the synthesis of g-C3N4 and SrZrO3 heterostructures showing photocatalytic H2 evolution. Below are some of the comments that authors should look into improving the manuscript.

1.       The authors have mentioned in section 2.4, “it was irradiated by a 1000 W Xe lamp with an optical filter (cutoff of 420 nm) or 500 W 89 Hg lamp for 8 h at room temperature under constant stirring”. Which method (1000 W Xe or 500 W 89 Hg) did the authors use should be exactly mentioned?

2.       It can be noticed from the XRD and TEM image SrZrO3 almost show crystalline structure, and adding 10:1 of SrZrO3: g-C3N4 doesn’t show any XRD peak of SrZrO3 in the heterostructure. What is the reason? The authors should recheck if the ratio of addition is correct.

3.       The lattice fringes shown in Figure S1 seems similar as it belongs to the SrZrO3 121 plane. Authors should reanalyse and specify the exact plane it belongs.

4.       It can be noticed from the XPS survey spectra there is no Nitrogen peak around 396 to 402 eV. How exactly would the authors mention the prepared material is g-C3N4 heterostructures? It would be better to repeat the XPS measurements.

5.       Likewise from XPS the authors have mentioned “the Two O 1s peaks centered at 529.4 eV and 531.5 eV were 126 ascribed to adsorbed oxygen and lattice oxygen”. How would you explain the bonding nature of SrZrO3?

6.       EDS mapping also shows very less amount of C and N. The authors should justify this in terms of XRD, TEM and XPS with details of the qualitative analysis.

7.       The authors have only used one ratio of heterostructures (10:1 of SrZrO3: g-C3N4) and confirmed it shows higher H2 evolution. It would be better to show different ratios of SrZrO3: g-C3N4 performance for a better understanding of the efficiency.

8.       The authors should provide more experimental evidence on the Photocatalytic mechanism of the heterostructures.

Reviewer 4 Report

Authors have reported the synthesis of g-C3N4/SrZrOand utilized them as photocatalysts for hydrogen evolution in UV and visible light photocatalysts. However, the explored material performs better in UV than visible light. Most importantly, the synthesized materials only act effectively in UV light. The results don't justify the meaning of the title of the manuscript. So, I don't recommend this work publication in this journal.